# Performance of a Prototype Boom Sprayer for Bed-Grown Carrots Based on Canopy Deposition Optimization, Ground Losses and Spray Drift Potential Mitigation in Semi-Field Conditions

Aude Lamare [1], Ingrid Zwertvaegher [2], David Nuyttens [2], Paolo Balsari [3], Paolo Marucco [3], Marco Grella [3], Amedeo Caffini [4], Nikos Mylonas [5], Spyros Fountas [5] and Jean-Paul Douzals [1,*]

[1] Institut National de Recherche pour l'Agriculture, l'Alimentation, et l'Environnement (INRAE), Université de Montpellier, 34196 Montpellier, France; aude.lamare@inrae.fr
[2] Flanders Research Institute for Agriculture, Fisheries and Food (ILVO), 9820 Merelbeke, Belgium; ingrid.zwertvaegher@ilvo.vlaanderen.be (I.Z.); david.nuyttens@ilvo.vlaanderen.be (D.N.)
[3] Department of Agricultural, Forest and Food Sciences (DiSAFA), University of Turin (UNITO), 10095 Grugliasco, Italy; paolo.balsari@unito.it (P.B.); paolo.marucco@unito.it (P.M.); marco.grella@unito.it (M.G.)
[4] Caffini Spa, 37050 Palu, Italy; amedeo.caffini@caffini.com
[5] Department of Natural Resources Development and Agricultural Engineering, Agricultural University of Athens (AUA), 11855 Athens, Greece; nmylonas@aua.gr (N.M.); sfountas@aua.gr (S.F.)
[*] Correspondence: jean-paul.douzals@inrae.fr; Tel.: +33-467166503

**Abstract:** The H2020-project OPTIMA concept of smart sprayer relies on several functionalities, including variable nozzle spacing for bed-grown carrots, based on an air-assisted boom sprayer. A prototype boom was designed and evaluated though canopy deposition, ground losses, and spray drift potential. Four bed spray configurations, including various nozzle types, angles, and sizes (XR8004, combination of AIUB8504/AI11004, AI8004, and XR8002) at the most appropriate nozzle spacing and height, were tested and compared to a broadcast application (XR11004). Deposition measurements were performed on carrots in bins at early and full-grown stages with respective target zone width of 1.4 m and 2.2 m. Spray drift potential measurements were performed following ISO 22401, 2015. The spray boom was equipped with an air sleeve providing different air speeds (0, 4, 8 m s$^{-1}$). The relative depositions at both growth stages showed a significant effect of spray configuration and lowest values were found for the broadcast application. The configurations consisting of air inclusion nozzles generated the lowest drift potential compared to the broadcast application, although not significantly different. Bed spray configurations can thus improve canopy depositions and spray drift potential compared to a conventional broadcast application when the boom height and the nozzle spacing are adjusted to the growth stage.

**Keywords:** bed-grown crop; air support; spray deposition; spray drift potential; smart sprayer; nozzle type

## 1. Introduction

In Europe, about 4.7 million tons carrots are produced, representing 103 thousand hectares [1]. Like many other vegetables, carrots represent an essential food component for humans, mostly for baby food. Minimizing plant protection product (PPP) residues is therefore a strong driver for limiting the use of PPPs [2]. In modern agriculture, PPPs are widely used to increase the yield and the quality of the harvest. For example, alternaria leaf blight (caused by Alternaria dauci) is one of the harmful diseases of carrot crop affecting yield. The sensitivity to the disease depends on fertilization, fungicide applications, and cultivars [3]. However, their use can cause environmental and sanitary problems through

direct operator exposure during spray application [4] or indirect exposure of residents, bystanders, and off-target areas via spray drift [5,6]. Ground losses may contaminate the soil, air, and water with adverse effects on plants and wildlife [7]. One way to mitigate the impact of PPP is to improve the efficiency of the spray application, i.e., to deposit the product on the target crop and to limit losses to the environment. That is one of the goals of the H2020-project OPTIMA (*OPTimised Integrated pest MAnagement for precise detection and control of plant diseases in perennial crops and open-field vegetables*, www.optima-h2020.eu (accessed on 1 April 2022)) in which a concept of smart sprayer includes several functionalities among which variable nozzle spacing, specifically for bed-grown carrots

Raised bed cropping is a traditional farming system for some open field vegetables such as carrots. Raised beds encourage the concept of controlled traffic [8], where all vehicle wheels travel along the furrows between the beds, thus limiting compaction of the soil on the majority of the field [9]. Taproot length and yield are generally negatively affected by soil compaction [10]. Carrots grow best on deep, loose, well-drained mineral and organic soils with good water holding [11]. They are therefore typically grown on sandy-loam soils that are deeply cultivated to reduce compaction. However, when those ideal soil types are not available, farmers are often advised to plant carrots on raised beds [12]. In addition, raised beds are a primary tool for field drainage [9,13], and can thus improve water relations in the root zone, possibly resulting in longer carrots [12], whereas in more arid parts of the world, raised beds are successful in improving the efficiency, and thus sustainability, of irrigation of arable crops, such as maize and wheat [14,15].

As for other row crops, these bed-raised vegetables are generally sprayed against diseases using a conventional horizontal boom sprayer applying a broadcast application. Such a broadcast application generally does not consider any modification of the nozzle spacing or boom height with regard to the raised bed configuration and developmental changes in canopy width and density related to the growth stage. Broadcast applications can thus result in substantial losses of product between beds, especially during the early stages of the crop growth. Indeed, adjusting the width of the spray pattern to match the width of the target plants has been shown to have economic benefits by saving products, as well as resulting in better deposition efficiency compared to a broadcast application. For example, a precision band spraying system using vision sensors and rotating nozzles was studied [16]. The nozzle rotation allowed the spray band to be adjusted to match the target width. The authors demonstrated a 66 to 80% reduction in spray application rates, a 2.5 to 3.7 times increase in target deposition efficiency, and a 72 to 90% reduction in non-target deposition on the soil surface compared to conventional spraying. In addition, the reduction in airborne displacement of spray liquid [16] suggests a considerable decrease in spray drift with such precision spraying techniques.

Another way to optimise spray applications is to adapt the nozzle spacing and boom height to the bed width. This strategy allows to better limit the spray liquid to the targeted crop and to minimise losses between the beds. Spray patterns for different bed widths and the ability to apply different dose rates depending on the crop canopy density were studied [17–19] However, these studies did not consider the possibility of adjusting nozzle spacing and boom height to the canopy width that increases during the growing season. The benefit of spray configurations with adjustable nozzle spacing and height is still to be demonstrated in terms of canopy deposition, ground losses, and spray drift, possibly further reducing PPP use and environmental contamination, as was the goal in this study.

To select the most optimal sprayer configurations and their most appropriate nozzle spacing and height for different target zone widths, which depends on the growth stage of the bed-grown carrots, preliminary spray distribution measurements were carried out and compared to a reference broadcast application [20]. Four bed spray configurations were identified for spraying different target zone widths (ranging from 1.2 to 2.2 m) with high uniformity (CV < 12%) and minimal losses out of the target zone (<17%), when applied at the most appropriate nozzle spacing and height (varying from 0.35 to 0.65 m). The objective of the present study is to validate these four bed spray configurations under

semi-field conditions for (i) spray deposition on early and full-grown carrots and (ii) spray drift potential using a drift test bench according to ISO 22401 (2015), by comparing them with a broadcast application.

## 2. Materials and Methods

### 2.1. Spray Boom Equipment

A prototype boom was designed at INRAE (Montpellier, France) using a short spray boom section with an air sleeve provided by Caffini S.p.a. (Palù, Italy). The boom was 1.96 m wide and equipped with four nozzles (Figure 1).

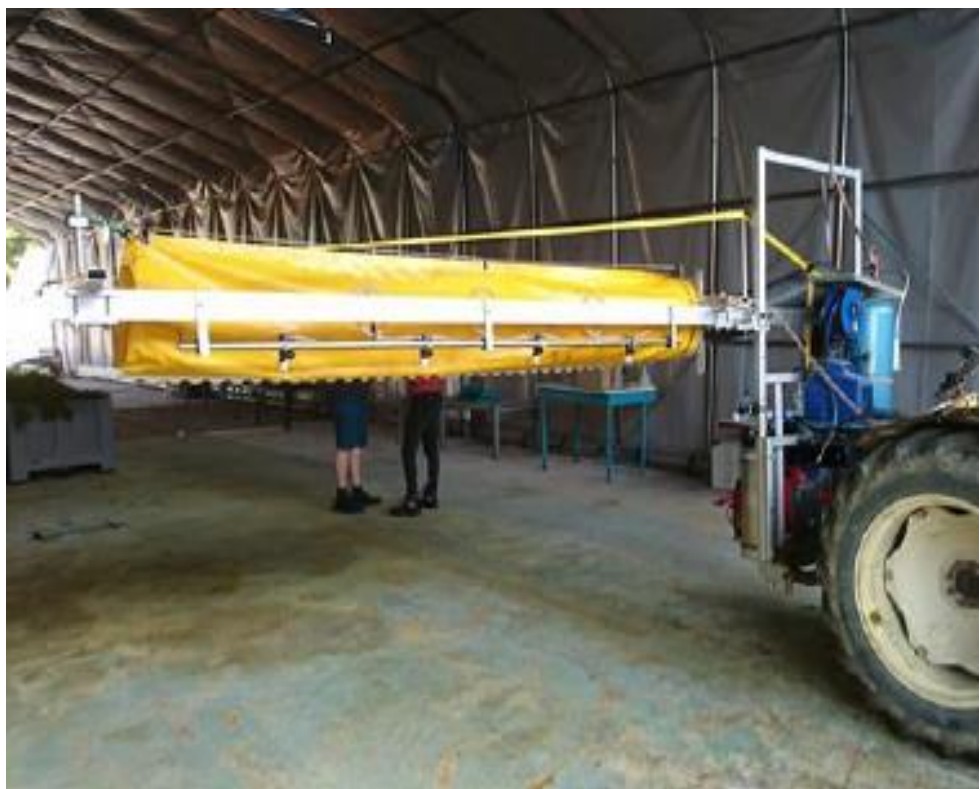

**Figure 1.** Prototype boom fitted with four nozzles.

Contrary to a conventional boom, the nozzle holders were mounted to a support that could slide along the boom to the desired position, allowing a variable spacing between adjacent nozzles ranging from 0.35 m to 0.60 m. In order to adjust the boom height, the boom was mounted to a metallic chassis fixed to the 3-point hitch of the tractor. A stand-alone electrical generator (240 V, Honda Inverter EU 20i, Honda, France) provided energy to both the air compressor (Lacme, Airlane 12, France) that pressurised the spray liquid tank through a pressure controller, and to the frequency controller (Siemens Sinamics G110, Germany) connected to a three-phase electrical motor and fan (Sodeca, MSE390S, Girona, Spain) that supplied the air support system.

### 2.2. Sprayer Configurations and Spray Settings

Table 1 gives an overview of the different configurations and settings tested for spray deposition and spray drift potential. For both deposition and drift potential measurements, the reference was considered to be a broadcast application of 158 l ha$^{-1}$, using a fixed nozzle spacing and height of 0.5 m with XR110 04 flat fan nozzles (TeeJet, Spraying Systems Co., Wheaton, IL USA) at a spray pressure of 300 kPa, without air support, and operated at 12 km h$^{-1}$.

**Table 1.** Spray configurations and air support settings tested for spray deposition on full-grown and early-stage carrots and for potential drift measurements.

| Application | Nozzles (N) | Air Support (m s$^{-1}$) | Nozzle Height/Spacing (m/m) | | | Application Rate (l ha$^{-1}$) [a] |
| | | | Deposition on Full-Grown Carrots | Deposition on Early-Stage Carrots | Drift Potential | |
|---|---|---|---|---|---|---|
| Reference broadcast application | XR 110 04 (4) | 0 | 0.50/0.50 | 0.50/0.50 | 0.50/0.50 | 158 [b] |
| Bed spray application | XR 80 04 (4) | 0<br>4<br>8 | 0.60/0.60 | 0.40/0.40 | 0.60/0.60 | 135 [c] |
| | AI 80 04 (4) | 0<br>4<br>8 | 0.60/0.60 | 0.40/0.40 | 0.60/0.60 | 135 [c] |
| | XR 80 02 (4) | 0<br>4<br>8 | 0.60/0.60 | 0.40/0.40 | 0.60/0.60 | 68 [c] |
| | AIUB 85 04 (2) + AI 110 04 (2) | 0<br>4<br>8 | 0.60/0.60 | 0.45/0.45 | 0.60/0.60 | 135 [d] |

[a] Application rate expressed as l ha$^{-1}$ of total ground area applied at 12 km h$^{-1}$ forward speed and 300 kPa spray pressure. [b] Broadcast application with 42 nozzles on a 21 m spray boom. [c] Bed spray application with 36 nozzles (4 nozzles per bed) on a 21 m spray boom. [d] Bed spray application with 36 nozzles (4 nozzles per bed, incl. 2 off-center nozzles) on a 21 m spray boom.

The bed spray configurations were selected based on spray distribution measurements carried out and described in [20]. Spray distribution measurements were performed for 13 nozzle configurations consisting of different nozzle types (conventional flat fan XR, air inclusion (AI) flat fan, off-center nozzles), nozzle size (ISO 02, ISO 04), spray angle (80°, 110°), and number of nozzles (3 or 4 nozzles per bed). Per nozzle configuration, measurements were operated at an equal nozzle spacing and height ranging from 0.35 to 0.65 m. For six different target zone widths (ranging from 1.2 to 2.2 m with intervals of 0.2 m), the coefficient of variation (CV) in the target zone and the percentage of losses outside the target zone (%) of each configuration were determined. Four configurations, each consisting of four nozzles, met the criteria of low variability within the target zone (CV < 12%) and minimal losses out of the target zone (<17%) at the different target zone widths, when applied at the most appropriate nozzle spacing and height. These bed spray configurations consisted of four TeeJet nozzles per configuration, i.e., XR8004/XR8004/XR8004/XR8004, XR8002/XR8002/XR8002/XR8002, AI8004/AI8004/AI8004/AI8004, AIUB8504/AI11004/AI11004/AIUB8504. All tests were performed at 300 kPa and 12 km h$^{-1}$, but depending on the configuration, the most appropriate nozzle spacing and boom height for the specific target zone width determined in [20] was used. The deposition measurements were carried out at two different growth stages, i.e., early-stage carrots and full-grown carrots, corresponding to respectively a target zone width of 1.4 m and 2.2 m. The boom height and nozzle spacing were adapted accordingly (0.4 or 0.45 m for the early-stage carrots and 0.6 m for the full-grown carrots). For drift potential measurements, 0.6 m boom height and nozzle spacing were used for the bed spray configurations to simulate the highest risk for drift generation. The spray deposition and drift potential of the bed spray configurations were tested at three different air support settings, i.e., no air support, air velocity of 4.0 m s$^{-1}$ and air velocity of 8.0 m s$^{-1}$ measured at the outlet of the air sleeve. In total, 13 settings were tested (Table 1). Each setting was replicated 3 times for deposition on carrots and spray drift potential. All experiments were achieved under a tunnel structure (Figure 1) of 24 m × 8 m in order to limit the effect of wind conditions. Air temperature and humidity were measured under the tunnel at a height of 2 m above ground using a Vaisala HSM 110 sensor (Vantaa, Finland) directly

connected to a computer. Table 2 presents the meteorological conditions (average ± s.d.) during a repetition from the beginning of the spray up to the total collection of collectors (duration of about 15 min in average, collection frequency 5 Hz) representing about 180 points per replicate.

**Table 2.** Meteorological conditions during deposition and spray drift potential measurements.

|  | Spray Configuration [a] | Temperature (°C) Mean ± s.d. | Relative Humidity (%) Mean ± s.d. |
|---|---|---|---|
| Deposition Full-grown carrots | XR 110 04 | 18.4 ± 2.9 | 81.0 ± 12.7 |
|  | AI 80 04 | 15.5 ± 0.5 | 51.0 ± 2.8 |
|  | XR 80 04 | 18.0 ± 1.1 | 68.8 ± 5.9 |
|  | XR 80 02 | 11.6 ± 1.4 | 87.5 ± 7.9 |
|  | AIUB 85 04/AI 110 04 | 10.2 ± 1.5 | 68.3 ± 6.8 |
| Deposition Early-stage carrots | XR 110 04 | 13.1 ± 0.2 | 53.0 ± 1.0 |
|  | AI 80 04 | 13.5 ± 0.3 | 51.7 ± 2.5 |
|  | XR 80 04 | 14.2 ± 0.6 | 46.5 ± 3.8 |
|  | XR 80 02 | 17.9 ± 0.8 | 53.5 ± 2.8 |
|  | AIUB 85 04/AI 110 04 | 21.2 ± 1.5 | 46.2 ± 4.1 |
| Spray Drift Potential | XR 110 04 | 16.7 ± 0.1 | 93.0 ± 0.1 |
|  | AI 80 04 | 17.1 ± 0.2 | 80.5 ± 3.5 |
|  | XR 80 04 | 19.7 ± 0.3 | 65.5 ± 3.8 |
|  | XR 80 02 | 16.2 ± 0.2 | 92.3 ± 1.2 |
|  | AIUB 85 04/AI 110 04 | 18.7 ± 0.6 | 68.3 ± 4.7 |

[a] See Table 1.

*2.3. Spray Deposition*

2.3.1. Crop Characteristics

Two cultivars, namely Maestro® and Soprano® (Vilmorin, Paris, France), were selected to represent two different crop structures with either a spread or upright canopy configuration, respectively. The carrots were seeded (early-stage carrots-BBCH15–5 leaves) or planted (full-grown carrots-BBCH 49, ready to harvest) in bins of 1.0 m × 1.2 m, according to the plantation structure of 3 lines × 3 rows per bed, with 0.4 m between rows, and 50–60 mm between lines, as described in [20].

This plantation scheme resulted in approximately 150 plants per m$^2$, representing a realistic crop density of 1.5 million plants per hectare. The bins were filled with a breeding ground up to 0.16 m from the top of the bins. The full-grown carrots came directly from a carrot growing estate (Planète Végétale estate, Cestas, France) and were replanted in 6 bins (3 lines × 3 rows, Figure 2a) the day after harvest. The deposition measurements were performed the next days. Carrots for the early-stage experiment were seeded in 3 bins (3 lines × 2 rows, Figure 2b) at INRAE Montpellier facility in September 2019 and used for experiments three months later. Five plants were defoliated, and leaves were scanned to determine the leaf area index. Average LAI values of 2.3 ± 0.4 and 4.6 ± 1.2 were found for the early-stage and full-grown carrots, respectively.

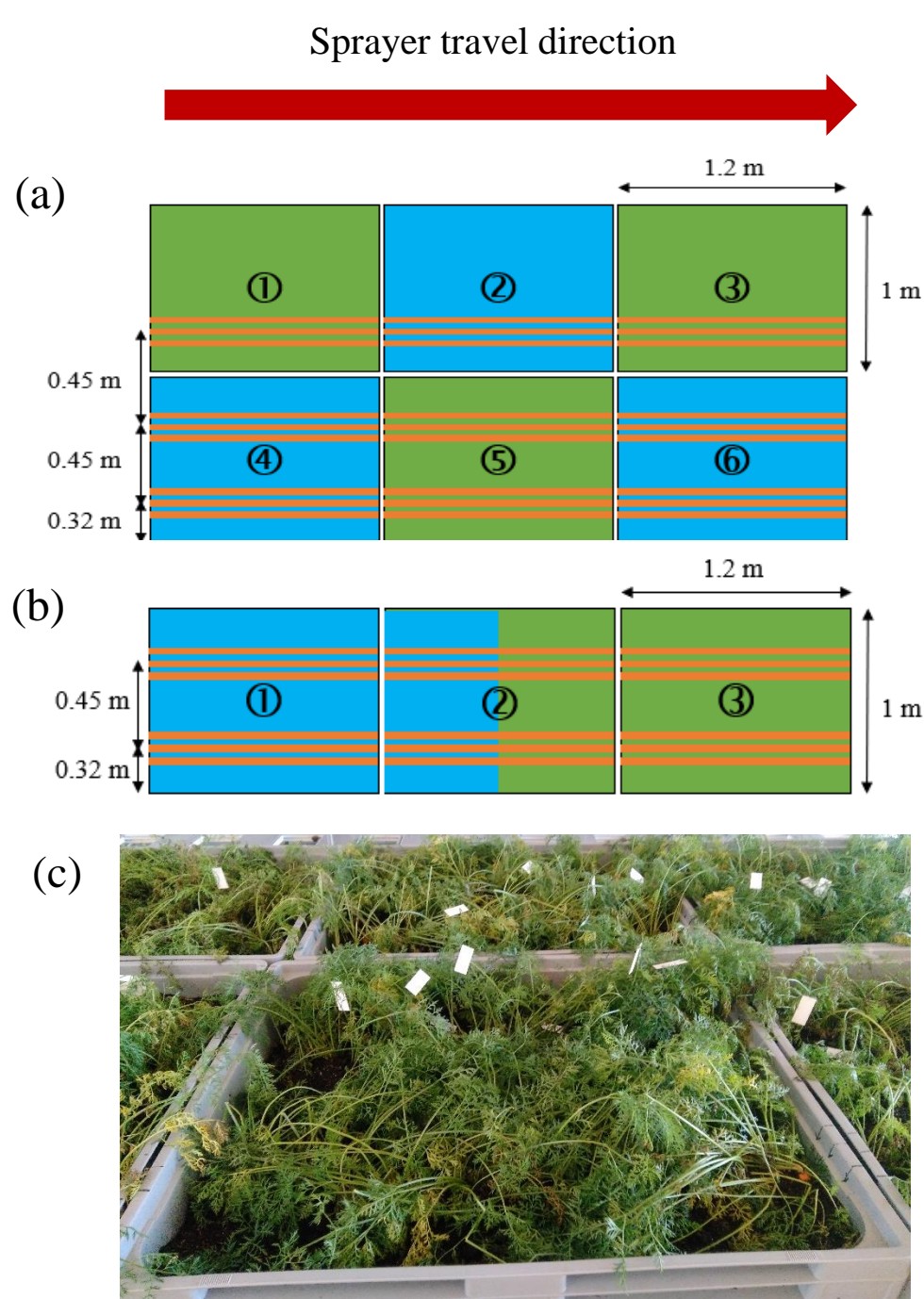

**Figure 2.** Plantation scheme in bin. Full development stage (**a**), early growth stage (**b**), deposition collectors in the vegetation (**c**).

### 2.3.2. Spray Deposition Assessment

Plastic collectors (dimension 70 mm × 30 mm, thickness 0.3 mm; ITW Formex, Carol Stream, IL, USA) were used to assess the canopy deposition and ground losses (Figure 2c). On full-grown carrots, the collectors were placed in the canopy with either a horizontal or vertical position to evaluate the canopy deposition, and horizontally on the ground, preferably under the leaves, to evaluate the ground losses. Six collectors were placed per zone (ground or canopy) and per bin, corresponding to 72 collectors per spray experiment (2 zones × 6 collectors × 6 bins). The collectors on the ground were placed in between the lines to avoid a shielding effect from the edges. As the top of the canopy was higher than the edges, this was not done for the collectors in the canopy. With the early-stage

carrots, it was not possible to sample the spray losses to the ground under the canopy due to the low crop height. Therefore, only four collectors were placed in the canopy per bin, corresponding to 12 collectors per spray experiment (1 zone × 4 collectors × 3 bins). Collectors were placed randomly but 0.3 m away from the bin edges to avoid border effects.

Before placing the new collectors, the canopy was allowed to dry completely (the drying time depended on the temperature and the wind). Since all trials were performed on the same plots, the risk of cross contamination from one trial to another was evaluated. For this purpose, a set of blank collectors was placed in the canopy between two consecutive experiments for some time (10 min). After analysis, these collectors showed no measurable contamination.

Spray mixture of E102 Tartrazine yellow food dye 85% *w/w* (Alpasud, France) with water, at a concentration of 5 g L$^{-1}$, were used in the spray deposition experiments. A volume of about 50 mL of spray mixture was systematically sampled at a nozzle outlet before each spray application to determine the mixture concentration. After each repetition, the collectors from the same bin and the same zone (canopy or ground) were placed into a small container of 100 mL, washed with 20 mL of deionised water for the canopy collectors or 10 mL for the ground collectors. A sample of each washing solution was placed in a 4 mL cuvette. The Tartrazine concentration of every sample was determined by measuring the absorbance of the washing solution with a spectrometer set at a wavelength of 427 nm (Secomam, Uviline 9100, Champigny sur Marne, France) and by comparing the results against the calibration curve. In case the concentration of the dye was too high, samples were diluted. Deionised water was used as a blank to calibrate the spectrometer. Spray mix samples were used to draw the calibration curves considering successive dilutions up to the upper detection limit of quantification of the spectrometer. Three sub-diluted samples were measured to draw the calibration curve. The absolute spray depositions were calculated using Equation (1) according to ISO22401 (2015):

$$D_i = \frac{(\rho_s - \rho_b) \times V_{dil}}{\rho_{spray} \times n_{coll} \times A_{coll}} \tag{1}$$

where $D_i$ is the absolute spray deposition per sample (µL cm$^{-2}$), $\rho_s$ is the absorbance value of the sample (adim.), $\rho_b$ is the absorbance value of the blank (adim.), $V_{dil}$ is the volume of the dilution liquid (deionised water) added per sample to extract the tracer deposition (ml), $\rho_{spray}$ is the estimated absorbance value of the spray mix concentration applied during the test and sampled at the nozzle outlet (adim.), $n_{coll}$ is the number of collectors per sample (adim.), $A_{coll}$ is the surface area of one side of the collector (cm$^2$).

*2.4. Spray Drift Potential*

2.4.1. Experimental Drift test Bench Setup

The spray drift potential was determined using a field drift test bench in accordance with ISO 22401 (2015). Details about the functioning of the bench are described in [21].

The test bench consisted of a 12.0 m × 0.5 m steel frame (Figure 3a) with 22 slots situated at intervals of 0.5 m (Figure 3b), in which Petri dishes of 80 mm diameter (6300 mm$^2$) were placed to catch droplets in suspension in the air at different times right after the spray application. The slots were equipped with sliding metallic covers, which completely covered the slots during spraying, thus safeguarding the Petri dish collectors from unintended spray deposition. The slots were simultaneously opened when the actuator was activated by passing of the spray boom. The actuator consisted of a vertical pole that was positioned 2.0 m ± 0.01 m after the center of the last slot (Figure 3). The last two slots were always uncovered, thus allowing to estimate the effective application rate from the direct deposition on the collectors. A minimum time of 60 s was given for droplet sedimentation on the Petri dishes after the opening of the slots.

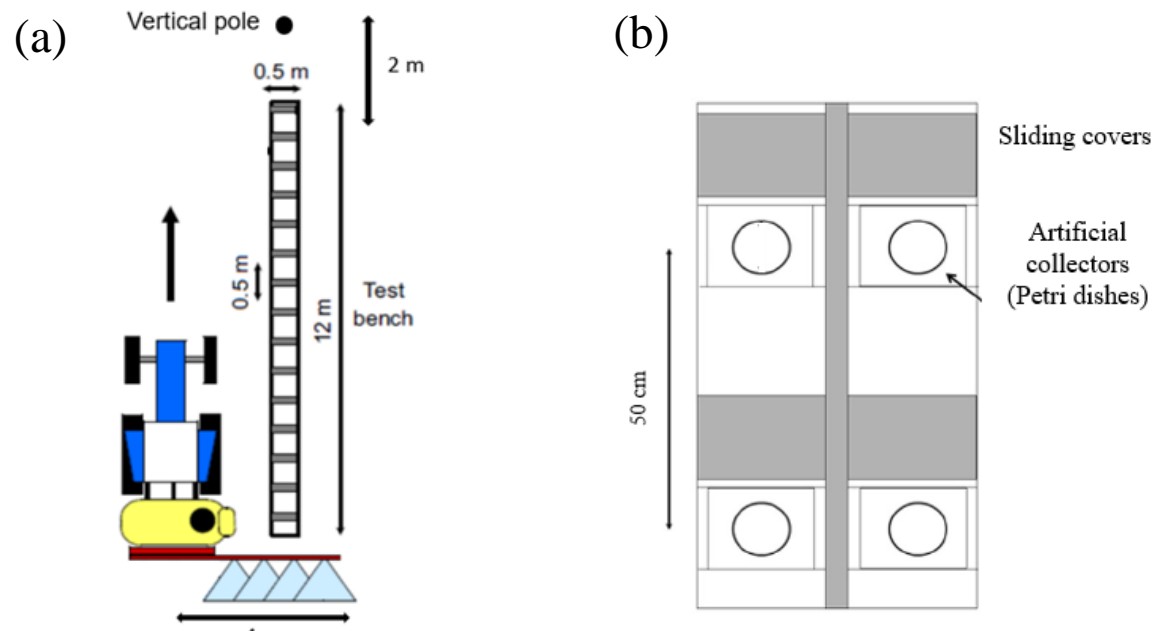

**Figure 3.** Potential drift test bench (**a**), close-up on slots for artificial collectors (**b**).

### 2.5. Drift Potential Value (DPV)

As for the deposition experiments, a spray mixture of Tartrazine at a concentration of 5 g l$^{-L}$ in water was used in the spray drift potential trials. A volume of about 50 mL of spray mix was systematically sampled at a nozzle outlet before each spray application in order to determine the spray mix concentration and to draw calibration curves (cf Section 2.3.2). After each trial, the Petri dishes were collected, washed with 10 mL of deionised water and placed in an orbital agitator to be shaken for 10 min. The Tartrazine concentration of every sample was determined by measuring the absorbance of the washing solution with a spectrometer (cf. Section 2.3.2). For each individual test, the deposition (expressed in μL cm$^{-2}$) on each Petri dish collector ($D_i$) was calculated using equation (1). From the absolute depositions of the 20 covered collectors, the Drift Potential Value (DPV) was calculated using Equation (2) according to ISO22401 (2015):

$$\text{DPV} = \frac{\sum_1^{20} D_i}{RSD} * 100 \tag{2}$$

where DPV is the drift potential value (adim.), $D_i$ is the spray deposition on each collector positioned in the covered slots (μL cm$^{-2}$) calculated from the absorbance values and the calibration coefficient (cf. Section 2.3.2), and RSD (Reference Spray Deposit) is the theoretical amount of spray deposit in a treated area with a given application rate (μL cm$^{-2}$). The *RSD* is equal to the application rate given in Table 1 but expressed as μL cm$^{-2}$. The absolute deposition on the uncovered Petri dishes was used to estimate the effective application rate.

### 2.6. Statistical Analysis

To account for the different spray applications used, relative canopy depositions and ground losses were calculated as percentage of the theoretical application rates (Table 1). Statistical analyses were performed using R (version 1.2.5033). Three-way ANOVAs with *Spray configuration*, *Air support setting*, *Cultivar*, and their interactions as fixed factors were performed on the relative canopy deposition in full-grown carrots, the relative ground losses in full-grown carrots, and the relative canopy deposition in early-stage carrots (dependent variables). The final models were selected using backward selection. Step by step, the interactions and then the independent variables were tested and removed when not significant ($p > 0.05$). The final models were evaluated by checking the normal

distribution and homoscedasticity of the residuals. If these two criteria were valid, Tuckey's post-hoc tests were performed. Otherwise, non-parametric Kruskal–Wallis and pairwise Wilcoxson tests were used to analyze the significant variables. Non-parametric tests were performed on relative canopy deposition and ground losses in full-grown carrots and parametric tests were performed on relative canopy depositions in early-stage carrots.

Concerning spray drift potential, Drift Potential Value data were transformed using ln(DPV) transformation to reach homoscedasticity and residuals normality, prior to the statistical analysis. Statistical differences among ln(DPV) values were evaluated using a two-way ANOVA including *Spray configuration* and *Air support setting* as independent variables. As for the deposition measurements, a backward selection was used. Post hoc comparisons were carried out using Tukey test.

## 3. Results

### 3.1. Canopy Deposition on Full-Grown Carrots

Spray configuration showed a significant effect on the relative canopy deposition [$F_{(4, 229)} = 21.831$, $p = 2.845 \times 10^{-15}$] in full-grown carrots, while *Air support setting* [$F_{(2, 231)} = 2.3$, $p = 0.105$] and *Cultivar* [$F_{(1, 232)} = 0.003$, $p = 0.957$] had no significant effect. Figure 4a shows the mean relative canopy deposition per spray configuration. The reference broadcast configuration XR11004 resulted in a relative canopy deposition of 45%. All bed spray configurations except XR8004, had a significantly higher relative canopy deposition than the reference. Configuration AIUB8504/AI11004 had the highest relative canopy deposition (67%), followed by XR8002 (59%), and AI8004 (56%). Configuration XR8004 had the lowest relative canopy deposition (49%), slightly higher but not statistically different from the reference broadcast application. The bed spray configurations thus resulted in (slightly) higher relative canopy depositions on full-grown carrots, although the boom height of the bed spray configurations was higher than that of the reference broadcast application (0.6 m vs. 0.5 m). Increasing the boom height generally leads to a decrease in deposition [22], because the droplets have more distance to travel before reaching the crop and are more likely to drift.

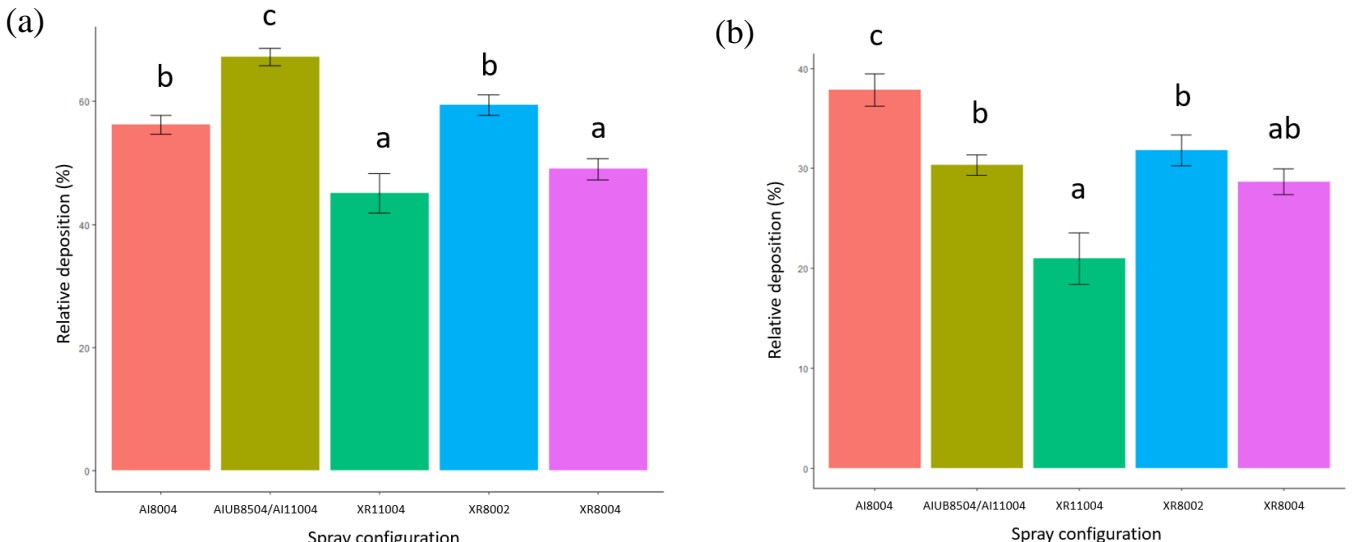

**Figure 4.** (**a**) Relative canopy deposition (%) and (**b**) relative ground losses (%) on full-grown carrots per spray configuration (mean ± standard deviation of the mean). Different letters denote significant differences between spray configurations (Pairwise Wilcoxon test, $p < 0.05$).

The canopy depositions obtained with the different configurations are thus a result of interacting factors, including different application rates, boom heights, nozzle spacings, nozzle types, and nozzle sizes. For example, similar relative depositions were obtained with

configuration AI8004 and XR8002, although at 300 kPa spray pressure these configurations produced fairly different droplet size spectra with respectively a very coarse and a medium spray (according to the BCPC spray quality class [23], as reported in [20]). Coarser droplets are generally less susceptible to air currents and are therefore more likely to collide with the plant surface as they are less likely to deviate from their initial path. By contrast, very small droplets follow almost exactly the streamlines of air flowing around an encountered object [24]. In addition, because of their higher velocity, coarse droplets are exposed to the influence of air movements for a shorter period than fine droplets [25]. Indeed, an increase in relative retention was found with droplet size spectrum on filter paper collectors [26], with highest retentions for the air inclusion nozzle compared to the flat fan nozzle.

Surprisingly, *Air support setting* did not significantly affect the relative canopy deposition in full-grown carrots. Air support is generally used to enhance transport of droplets to and into the canopy, thus improving spray penetration, deposition, and coverage, including at the underside of the leaves [27–32]. In this trial, the air support might not have been high enough to see a significant effect on canopy deposition. The cultivation in bin may also affect the influence of the air support since the boom was relatively high compared to the ground (0.8 m bin height, 1.2–1.4 m boom height), potentially affecting the airflow behavior. Additional measurements showed that air velocities of 4.0 and 8.0 m s$^{-1}$ at the outlet of the air sleeve resulted in air velocities of respectively 1.2 and 2.6 m s$^{-1}$ at 0.5 m below the outlet, i.e., more or less at canopy height. The air velocities obtained at canopy level were therefore lower than in [33] where the air velocity at canopy surface ranged between 5 and 15 m s$^{-1}$. However, the comparison with other studies is also possible considering the air velocity at the exit of the air sleeve of 8 m s$^{-1}$ [34], 6 to 18 m s$^{-1}$ [19] and up to 35 m s$^{-1}$ [35,36]. An increase in air velocity or a decrease in forward speed might positively affect canopy deposition. Indeed, an increase in droplet density, expressed as number of droplets per cm$^2$, on the leaves followed the increase in air flow velocity from 5 to 15 m s$^{-1}$ on a simulated crop canopy [33]. Furthermore, lower forward speeds also resulted in higher depositions [19,27] and droplet density [33] on artificial targets. However, in all previous cases, the travel speed varied from 3 to 8 km h$^{-1}$ that appeared lower than 12 km h$^{-1}$ as in the present study.

*Cultivar* had no significant effect on relative canopy deposition in full-grown carrots [F(4, 567) = 21.834, *p* = 2111]. This might be explained by both cultivars having a dense canopy at this stage, despite having different crop structures.

### 3.2. Ground Losses on Full-Grown Carrots

*Spray configuration* also had a significant effect on the relative ground losses [F(4, 229) = 11.19, *p* = 2.621 × 10$^{-8}$] in full-grown carrots, whereas *Air support setting* [F(2, 231) = 1.12, *p* = 0.33] and *Cultivar* [F(1, 232) = 5.048, *p* = 0.065] had no significant effect. The relative ground losses in full-grown carrots are shown in Figure 4b. In total, 21% of the theoretically applied volume was lost to the ground with the reference broadcast application (XR11004). As for relative canopy deposition, all bed spray configurations, except XR8004, had significantly higher relative ground losses than the reference broadcast application. Configuration AI8004 had the highest relative ground losses (38%), followed by configurations XR8002 (32%), AIUB8504/AI11004 (30%), and XR8004 (29%). The higher relative ground losses of the bed spray configurations can easily be explained by higher spray volumes being focused to the bed with these configurations. In comparison, the total spray volume applied with the broadcast application is distributed over the bed as well as between the beds, thus resulting in lower losses on the ground below the canopy. Much lower ground losses are expected between the beds with the bed spray configurations compared to the broadcast application. The coarser droplets produced by the air inclusion nozzles [20], which are more likely to bounce or shatter of the leaves, might also slightly explain higher ground losses with these configurations. Indeed, the largest proportions of shatter and the lowest proportions of adhesion and rebound were found with the air inclusion nozzle AI11008, in a study otherwise comprising of standard flat fan nozzles XR11001, XR11004 and XR11008 [26]. However, this does not explain the significant difference between configuration AI8004 and the

other bed spray configurations, which showed similar ground losses, indicating that a variety of factors is involved.

As for canopy deposition, relatively low air velocities and similar canopy density of both cultivars might explain the lack of a significant effect of *Air support setting* and *Cultivar* on the ground losses. Other authors, such as [28,30], reported higher ground losses with air-assistance in a potato crop, both at the ridges and the furrows. Although *Air support setting* was found to have no significant effect in this study, it might have influenced the ground losses to some extent, possibly explaining the higher losses with the bed spray applications (which included measurements without air support as well as with high and low air support) compared to the broadcast application (which were carried out without air support).

### 3.3. Canopy Deposition on Early-Stage Carrots

As for full-grown carrots, *Spray configuration* was found to be the only factor with a significant effect on relative canopy deposition in early-stage carrots [F(4, 79) = 10.202, $p = 1.048 \times 10^{-6}$]. Figure 5 shows per spray configuration the mean relative canopy depositions on early-stage carrots.

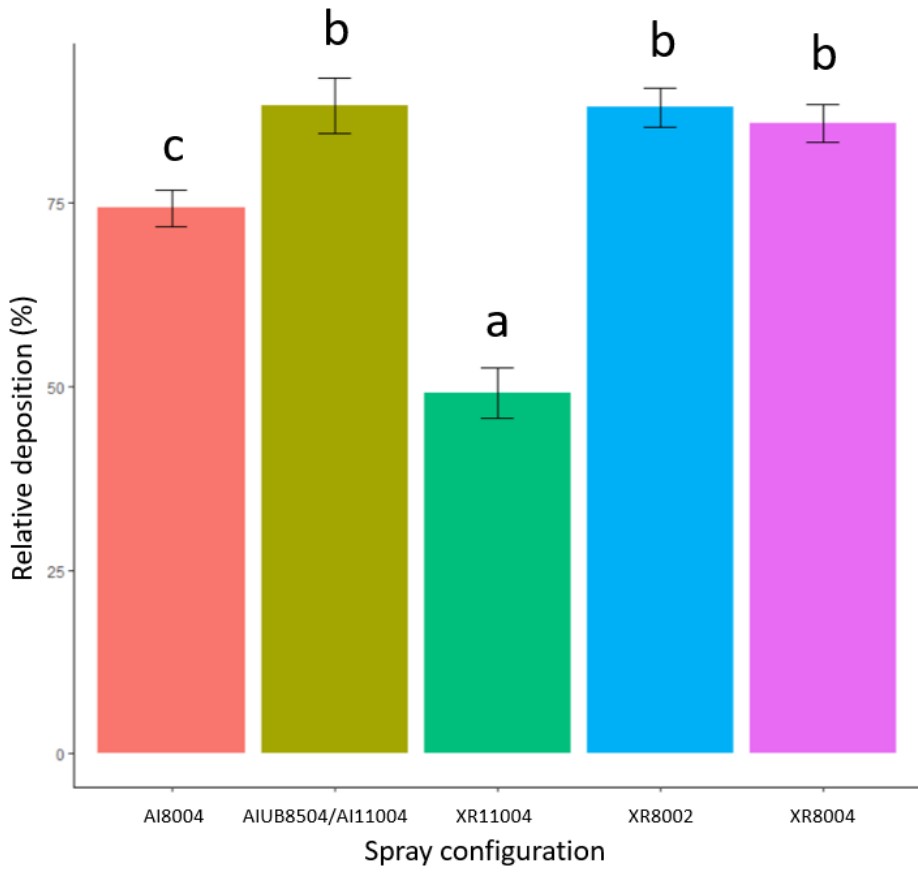

**Figure 5.** Relative canopy deposition (%) on early-stage carrots per spray configuration (mean ± standard deviation of the mean). Different letters denote significant differences between spray configurations (post hoc Tukey test, $p < 0.05$).

The relative canopy deposition of reference configuration XR11004 (49%) was significantly lower than that of the bed spray configurations. Configurations XR8002 and AIUB8504/AI11004 had the highest depositions (both 88%), closely followed by configuration XR8004 (86%), whereas configuration AI8004 showed the lowest deposition (74%). The relative canopy depositions of the bed spray configurations were not significantly different

from each other, except for configuration AI8004, which was significantly lower than the others, but still significantly higher than the reference broadcast application.

A similar relative canopy deposition on both early-stage and full-grown carrots was obtained with the reference broadcast configuration (around 45%). Both measurements were performed at 0.5 m nozzle height and spacing. For the bed spray configurations, the relative canopy depositions were found to be higher on the early-stage than on the full-grown carrots. This can be explained by the difference in the target zone width (1.4 m in early-stage carrots vs. 2.2 m in full-grown carrots). Relative canopy deposition was improved with the bed spray configurations compared to the reference broadcast application, proving that adjusting the configuration to the target zone width in combination with an adjusted nozzle height and spacing enhances the deposition on the canopy. *Air support setting* [F(2, 807) = 15.437, $p$ = 0.128] had no significant effect on the canopy deposition in early-stage carrots, most likely due to the low air velocity and specific experimental setup, as described above.

### 3.4. Spray Drift Potential

*Spray configuration* had a significant effect on ln(DPV) [F(4, 341) = 8.961, $p$ = 4.312 $\times$ 10$^{-5}$]. Figure 6 shows the mean DPV per spray configuration and the effect of spray configuration on ln (DPV).

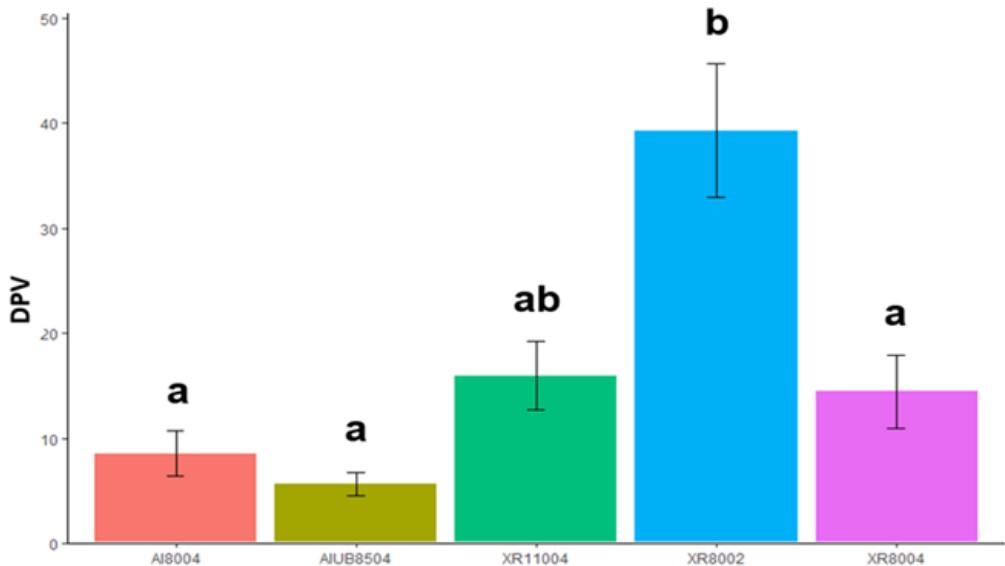

**Figure 6.** Drift Potential Value (DPV) of the different spray configurations (mean ± standard deviation of the mean). Different letters denote significant differences between spray configurations on ln(DPV) (post hoc Tukey test, $p$ < 0.05).

The ln(DPV) of the reference broadcast configuration XR11004 did not significantly differ from those of the bed spray configurations, although considerable differences in DPV and drift reductions were present. Configuration XR11004 had a DPV of 19, whereas XR8002, XR8004, AI8004 and AIUB8504 resulted in DPV's of 38, 15, 9, and 6, respectively and therefore in drift reductions of −97%, 24%, 55%, 70% compared to the reference broadcast application. As a general statement, the principle of the drift test bench is based on the evaluation of spray deposition on a narrow section under the spray boom, in this experimental set-up on the center of a bed. The spray drift potential of the bed spray configurations is therefore overestimated because, unlike broadcast applications, bed spray applications consist of 'full spray zones' and 'no spray zones'. Due to the width and position of the drift test bench in the experiments, only the 'full spray zone' was considered, and thus the lower spray drift potential in 'no spray zones' is not taken into account. Keeping this in mind, configuration XR8002 drastically increased the spray drift potential

compared to the reference. This can be explained by a combined effect of a higher boom height (0.6 m) and especially a finer droplet size. All else being equal, smaller nozzle sizes generally result in smaller droplet size spectra, and therefore droplets which are more prone to drift [37]. At 300 kPa spray pressure, nozzle XR8002 produced finer droplets with lower velocity than the reference nozzle XR11004 (VMD = 260 μm vs. 300 μm), as reported in [20], making them more prone to drift and explaining the increased DPV. In contrast, at 300 kPa spray pressure, air inclusion nozzles produced very coarse (VMD ranging from 443 μm to 460 μm) and therefore less driftable droplets [20]. The configurations using air inclusion nozzles thus resulted in the best drift reductions compared to the reference configuration, despite the higher boom height. These findings are in accordance with the drift test bench experiments performed by [21,38]. Similar results were found in spray drift field tests [39,40].

As for deposition measurement, *Air support setting* (F(2, 362) = 14.246, *p* = 0.683) had no significant effect on ln(DPV), possibly due to the relatively low air velocity as discussed above. Furthermore, the absence of vegetation during the drift test bench measurements might limit the use of these measurements in determining the effect of air support on spray drift potential, as typically an intercepting crop is needed to evaluate the use of air support. Various previous studies showed that air assistance reduced spray drift. For example, angling the air curtain can reduce drift of flat-fan nozzles over stubble by 60% [41]. Depending on the nozzle type [30], an additional drift reduction of 45 to 90% was obtained at 2 to 3 m from the last nozzle using air assistance in potato crops. A significant effect of air support on the total amount of spray drift for Hardi ISO F 110 02, F 110 03, and LD 110 02 nozzles by increasing droplet velocities [40] but contrasting results were also found [28]. More drift was produced with an air-assisted sprayer compared to a standard sprayer, although air-assistance resulted in higher biological efficacy on Brussel sprouts, despite being a difficult to spray crop. The authors thus concluded that air assistance can exacerbate drift unless the parameters such as air velocity and boom height are optimized. Several factors should therefore be considered when using air support.

## 4. Conclusions

Spray application efficiency is related to the capability to minimize spray losses to the ground and the air, while still guaranteeing a good level of spray coverage and deposition throughout the canopy. In this respect, a sprayer for bed-grown carrots, employing variable nozzle spacing and boom height to match the target zone width at different crop stages, was developed within the H2020-project OPTIMA. A first step was to select various nozzle types and configurations and the most optimal nozzle spacing and boom height. Four most promising bed spray configurations were tested in terms of spray deposition, ground losses and spray drift potential and compared to the reference broadcast application in semi-field conditions. The bed spray configurations resulted in higher relative canopy depositions and ground losses under the canopy, both at early-stage and full-grown carrots, due to relatively more spray being directed to the target zone compared to the broadcast application. Spray drift potential was closely related to nozzle type and droplet size spectrum, with increased drift potential for fine sprays and large drift potential reductions with air inclusion nozzles, even despite higher spray boom height than the reference broadcast application. It can be concluded that bed spray configurations can improve canopy depositions and limit spray drift potential compared to a conventional broadcast application, when the boom height and the nozzle spacing are adjusted according to the growth stage. Especially the configuration with air inclusion nozzles in combination with off-center nozzles, i.e., AIUB8504, resulted in high canopy depositions and drift reduction potential. The use of air support had no significant effect on spray deposition or spray drift potential, probably because of the relatively low airflow velocities. In a next step, the use of variable nozzle spacing and air support will be studied under real field conditions using the most promising configurations.

**Author Contributions:** Conceptualization, A.L., J.-P.D. and I.Z.; methodology, A.L. and J.-P.D.; validation, formal analysis, I.Z., D.N. and M.G.; investigation, A.L. and J.-P.D.; resources, A.C.; writing—original draft preparation, A.L. and J.-P.D.; writing—review and editing, I.Z., P.B., P.M., M.G., N.M., S.F. and D.N.; project administration, N.M. and S.F. All authors have read and agreed to the published version of the manuscript.

**Funding:** This research was funded by European Union's Horizon 2020 research and innovation program under grant agreement number 773718 (OPTIMA-project).

**Institutional Review Board Statement:** Not applicable.

**Informed Consent Statement:** Not applicable.

**Data Availability Statement:** The data presented in this study are available upon request from the corresponding author. Data will become openly available in a public repository that issues datasets with DOIs (i.e., Zenodo platform) in the following weeks, in line with the Horizon 2020 project requirements.

**Acknowledgments:** Planète Végétal SCEA Le Pot au Pin, Cestas 30610, France, Invenio, Ychoux 40160, France, AAMS BV, Madelgem, 9990; Belgium and Salvarani Srl, Poviglio (RE), Italy are acknowledged for the furniture of experimental material and practical support.

**Conflicts of Interest:** The authors declare no conflict of interest. The funders had no role in the design of the study; in the collection, analyses, or interpretation of data; in the writing of the manuscript, or in the decision to publish the results.

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
