# Peer review of "Performance of a Prototype Boom Sprayer for Bed-Grown Carrots Based on Canopy Deposition Optimization, Ground Losses and Spray Drift Potential Mitigation in Semi-Field Conditions"

_applsci, doi:10.3390/app12094462_

Round 1

Reviewer 1 Report

From my point of view, the manuscript is interesting and the research is conducted correctly.
However, there are small corrections to be made, which can improve the manuscript:
- bibliographic sources can be added to the introduction, as well as to the results.
- figure 1 and its title must be centered similar to figure 2
- tables 1 and 2 centered on the page
- figure 2 should be resized to be on a single page including its title.
- to check and delete blank lines, where applicable (ex.482-483)
- it is obligatory to check the bibliographic sources because 37 references appear in the References and only 36 appear in the text, and if the bibliography to be revised is restored, the order in which bibliographic sources are cited in the text, after the reference [16] line 173 follows the reference [33] line 241, this can be corrected.

Author Response

Dear Reviewer,

We thank you for your comments that were taken into consideration in the corrected version.

The manuscript presents a prototype boom, designed to evaluate variable nozzle spacing for bed-

grown carrots, based on an air-assisted boom sprayer. The manuscript studies ground losses and

spray drift potential in bed-grown carrots. Four bed spray configurations with various nozzle types,

angles and sizes were experimentally compared to a broadcast appliction, in order to find the best

nozzle spacing and height.

We would like to suggest a few modifications in the manuscript, as follows:

1) In the abstract it is mentioned the development of a “smart sprayer”. However, to be considered a

“smart sprayer”, the system should be able to measure some parameter (height or density of leaves,

for example) and adjust the spray accordingly to the measured parameter. Since the developed

system does not perform in this way, it is necessary to remove the word “smart” from the abstract.

Ans: The smart sprayer is a global concept within the project that includes a number of functionalities  among those variable nozzle spacing. The abstract and one sentence of the introduction (lines 53-54) were modified accordingly. As specified in the title, this study focused on variable nozzle spacing benefits  and there is no reference to the smart sprayer afterwards in the text.  

2) It is not mandatory, but improving the look of Tables 1 and 2 would be nice for the general

presentation of the manuscript.

Ans : all tables and figures were centered

3) In Lines 145-146 there is a problem with the text format. Please correct it.

Ans: The format was corrected.

4) In Table 2, it is presented the average and std deviation of temperature and RH measurements.

Please indicate the number of measurements used to calculate these parameters.

Ans : the text as modified as in line 173-174: (duration of about 15 min in average, collection frequency 5 Hz) representing about 180 points per replicates.

5) In Fig.2, it would be better to invert (a) and (b), showing the early growth stage before the full

development stage.

Ans: yes it sounds logical when referring to crop phenology. However since full grown stage experiments are introduced first (because the protocol was more extensive with either canopy and ground collectors). Figure 2 follows the same logics.  

6) In Line 229:

“Where Di is the absolute spray deposition per sample (μl cm-²), ρs is the absorbance value of the

sample (adim.), ρb is the absorbance value of the blank (adim.), Vdil is the volume of the dilution

liquid (deionised water) added per sample to extract the tracer deposition (ml), ρspray...”

the symbols used for “ρ” in the text look like a capital Q... Is it possible to change it?

Ans: Yes it was a typo effect. The typo was corrected accordingly in line 232-234.

7) Do you have any data (besides the LAI) on the carrots characteristics from each bed (mass,

length, etc.) to see if, besides the clear benefits for the environment and the PPP operators health,

the carrots also benefit from this optimized sprayer? If yes, it would be interesting to mention it in

the manuscript.

Ans : This study was punctual and only carried with water and dye. As mentioned in the conclusion, field tests are planned to test different spray configurations against the disease severity and yield.   

Author Response

(The authors gave the same response as above.)
